# Aging, Rather than Genotype, Is the Principal Contributor to Differential Gene Expression Within Targeted Replacement APOE2, APOE3, and APOE4 Mouse Brain

**DOI:** 10.3390/brainsci15101117

**Published:** 2025-10-17

**Authors:** Amanda Labuza, Harshitha Pidikiti, Melissa J. Alldred, Kyrillos W. Ibrahim, Katherine Y. Peng, Jonathan Pasato, Adriana Heguy, Paul M. Mathews, Stephen D. Ginsberg

**Affiliations:** 1Center for Dementia Research, Nathan Kline Institute, Orangeburg, NY 10962, USA; labuzaheller@gmail.com (A.L.); harshitha.pidikiti@nki.rfmh.org (H.P.); melissa.alldred@nki.rfmh.org (M.J.A.); kyrillos.ibrahim@nki.rfmh.org (K.W.I.); katherine.peng@nyulangone.org (K.Y.P.); paul.mathews@nki.rfmh.org (P.M.M.); 2Department of Psychiatry, New York University Grossman School of Medicine, New York, NY 10016, USA; 3Genome Technology Center, New York University Grossman School of Medicine, New York, NY 10016, USA; adriana.heguy@nyulangone.org; 4Department of Pathology, New York University Grossman School of Medicine, New York, NY 10016, USA; 5NYU Neuroscience Institute, New York University Grossman School of Medicine, New York, NY 10016, USA; 6Department of Neuroscience & Physiology, New York University Grossman School of Medicine, New York, NY 10016, USA

**Keywords:** apolipoprotein E, aging, Alzheimer’s disease, neurodegeneration, mouse model, RNA sequencing

## Abstract

**Background:** Apolipoprotein E (APOE) is the strongest genetic risk determinant for late-onset Alzheimer’s disease (AD). The APOE3 allele is risk-neutral, the APOE4 allele increases the risk of developing AD, and the APOE2 allele is neuroprotective. **Methods**: We utilized RNA sequencing of hemi-brains from a mouse model homozygous for each of these humanized APOE alleles to study gene expression profiles between mice aged 12 months of age (MO) and 18 MO, independent of β-amyloid and tau pathology. **Results**: More than half of the differentially expressed genes (DEGs) within each genotype were shared with at least one other APOE allele, including 1610 DEGs that were shared across the three genotypes. These DEGs represent changes driven by aging rather than APOE genotype. Aging induced DEGs and biological pathways involving metabolism, synaptic function, and protein synthesis, among others. Alterations in these pathways were also identified by DEGs unique to APOE4, suggesting that the APOE4 allele drives the aging phenotype. In contrast, fewer pathways were identified from DEGs unique to APOE2 or APOE3. **Conclusions**: Transcriptomic results suggest that the most significant impact on brain-level expression changes in humanized APOE mice is aging and that APOE4 exacerbates this process. These in vivo findings within an established model system are consistent with brain aging being the greatest risk factor for AD and suggest that APOE4 expression promotes an aging phenotype in the brain that interacts with, and contributes to, aging-driven AD risk. Results reinforce the impact age and APOE allele contribute to AD and age-related neurodegeneration, and foster greater mechanistic understanding as well as inform therapeutic intervention.

## 1. Introduction

The apolipoprotein E gene (APOE) has three different polymorphic alleles in humans, termed APOE2, APOE3, and APOE4 [1,2]. These three alleles correspond to variations in the amino acids at residues 112 and 158 of the protein, with APOE3 having a cysteine at position 112 and an arginine at position 158, while APOE2 has cysteine and APOE4 has arginine at both sites, respectively [1,2]. The APOE3 allele is the most common, with an estimated 79% global population, with 8% and 14% having an APOE2 or APOE4 allele, respectively [3,4].

The APOE gene has been identified as the strongest genetic risk factor for late-onset Alzheimer’s disease (AD), including in recent, large-scale, genome-wide association studies (GWAS), GWAS meta-analyses, and population-based meta-analyses [5,6,7,8]. APOE4 increases the risk of AD [5,8,9,10], while APOE2 may be protective against AD [8,11,12], and APOE3 is neutral. A single copy of APOE4 leads to a ~4-fold greater likelihood of developing AD compared to APOE3/APOE3 carriers, while two copies of APOE4 result in a ~12-fold increase [13]. The risk of AD is greater for heterozygous female APOE4 carriers than heterozygous males, while APOE4 homozygosity carries an equal risk for both sexes [10,14,15]. APOE4 also worsens neuropathological changes in AD patients compared to APOE3, including severe cerebral amyloid angiopathy and increased tau pathology encompassing more extensive regions within the brain [16,17]. Several of these neuropathological changes are increased with APOE4 and can occur independently of cognitive decline [18,19,20,21]. A USA cohort of 5000 subjects confirmed that APOE2 homozygosity had both lower odds of developing AD and milder neuropathological changes than APOE2/3 or APOE3/3 carriers [22].

Independent of AD, APOE has been shown to play a role in neurodegeneration [23,24]. APOE4 has been associated with the severity of TDP-43 pathology [25,26], Lewy body disease (LBD) [27,28], tau-mediated neurodegeneration [29], and metabolic syndrome (MetS) [30]. APOE mouse models demonstrate APOE4 expression alters the neuronal endosomal–lysosomal system [31,32]. These APOE4 effects are consistent with APOE4 driving aging-dependent changes in diverse cellular pathways [23,33,34,35]. In contrast to APOE4, APOE2 expression is protective against LBD [27] and correlates with longevity, independent of AD pathology [36,37].

The strong correlation between the APOE allele and neuropathological changes has led to several therapies targeting APOE, summarized by Serrano-Pozo [17,38], though none have been successful to date. It is therefore of interest to study APOE in the context of AD and Alzheimer’s disease-related dementias (ADRDs), along with the upstream and downstream pathways that APOE acts upon.

Multi-’omics have been used to study mechanisms of APOE alleles in neuronal health [31,32,35,39]. Our collaborative lab group reported brain mRNA expression differences driven by APOE genotype are related to the endosomal–lysosomal system in humanized APOE targeted replacement mice [31,32]. RNA sequencing (RNA-seq) of induced pluripotent stem cell (iPSC)-derived neurons, human iPSC (hiPSC)-derived glia, and cerebrospinal fluid (CSF) proteomics in human patients have revealed changes in synaptic function in APOE4 compared to APOE3 [40,41,42,43]. Since human studies intertwine genetic and environmental factors, including diet, lifestyle, aging, and AD pathology, we used a well-characterized in vivo humanized APOE knock-in mouse model comprised of targeted replacement ApoE2, ApoE3, and ApoE4 [32]. We combined brain-level RNA-seq with bioinformatic inquiry to interrogate changes in gene profiles due to aging in the context of APOE genotype. We hypothesize changes in gene expression will identify novel pathways to better understand the underlying mechanistic role(s) that APOE genotype plays in aging and AD dementia risk, which may help direct therapeutic intervention.

## 2. Materials and Methods

### 2.1. Animals

Mice used in this study were group-housed under controlled temperature and lighting conditions with same-sex littermates. Mice were given free access to food and water. Mice were originally developed on a C57Bl/6 background to express the human APOE gene in place of the mouse *Apoe* gene under the endogenous murine promoter [44]. This allowed for expression of human APOE at physiologically regulated levels in the same temporal and spatial pattern as endogenous murine *Apoe* [44]. APOE2, APOE3, and APOE4 refer to mice homozygous for either human APOE2, APOE3, or APOE4 (RRID:MGI:3695702, RRID:MGI:3695698, and RRID:MGI:4355228), respectively. Murine APOE genotype was confirmed using restriction fragment length polymorphism analysis [45]. The 12 months of age (MO) cohort had 3 male/3 female mice per genotype and the 18 MO cohort had 5 female mice per genotype.

### 2.2. RNA Sequencing

RNA was purified from the frozen left hemibrains of mice using the miRNeasy mini kit (Qiagen, Germantown, MD, USA). Right hemibrains were used for biochemistry as described previously [32,46]. Validation for quality and quantity was performed utilizing the RNA6000 Nano kit on an Agilent 2100 Bioanalyzer (Agilent, Santa Clara, CA, USA). RNA was stored at −80 °C prior to use. All RNA samples had an RNA integrity number (RIN) of 9 or higher. RNA-seq libraries were prepared on a Biomek Fxp liquid handler robot (Beckman, Indianapolis, IN, USA) using the TruSeq Stranded mRNA Library Prep kit (#20020595; Illumina, San Diego, CA, USA), starting with 500 ng of total RNA with 10 cycles of final PCR amplification. Equimolar pooling was performed on all libraries with single-read 50 base pair sequencing performed across three lanes of an Illumina HiSeq4000 flow cell at the New York University Grossman School of Medicine Genome Technology Center (NYUGSOM GTC).

### 2.3. Sequence Analysis

FASTQ files from the same biological sample with multiple lanes were merged prior to pre-processing. Since the lanes were technical replicates from the same run, they did not represent different batches [47,48,49]. Quality control of the raw reads was performed using FastQC (v0.11.9) [50]. Trimmomatic (v0.39) [51] was used to trim raw reads with any adapter contamination. This step was skipped if there was no adapter contamination. STAR aligner (v2.7.10a) [52] was used to generate a genome index and align trimmed reads to the indexed murine reference genome (GRCm39) in two-pass basic mode with default parameters [47,48,49].

### 2.4. Statistical Analysis and Bioinformatics

RSEM (v1.3.3) was used to quantify gene-level expression on aligned reads. Picard (v2.27.1) [53] was used to compute post-alignment and RNA-seq-specific quality metrics. Gene-level counts from RSEM [54] were used to perform differential gene expression analysis. Genes with over 0.1 counts per million in more than 40% of the samples were retained. TMM normalization was implemented by edgeR [55], accounting for any compositional differences between libraries. This step removes lowly expressed genes that provide little evidence of differential expression and increase statistical errors and false discovery rates [56,57].

Gene expression analysis was performed using the DREAM pipeline [58] built on top of the limma-voom pipeline from the variancePartition package [59]. In addition to genotype and age, the following variables were included as covariates: intergenic percentage, intronic percentage, and usable base percentage. Except for genotype and age, other covariates were computed from the RNA-seq reads by Picard. Differentially expressed genes (DEGs) were identified using the topTable function, with statistical significance defined as *p*-value < 0.05. DEGs were used for subsequent functional enrichment analyses.

Ingenuity Pathway Analysis (IPA; Qiagen, Germantown, MD, USA), Kyoto Encyclopedia of Genes and Genomes (KEGG) [60], and Gene Ontology (GO) bioinformatic pathway analyses were employed using DEGs, with *p* < 0.05 as the significance cutoff [47,48,49]. For input of the 1610 pan APOE DEGs in GO analysis, *p*-values from APOE4 DEGs were used as APOE4 DEGs had the highest significance of the three genotypes. Subdivisions of GO categories only included brain-related pathways in the “development” section, excluding pathways involved in peripheral organ development.

## 3. Results

The two greatest risk determinants for AD are aging and APOE genotype. We sought to examine the interface of these risk factors in the brains of mice homozygous for human APOE2, APOE3, or APOE4. These mice do not develop β-amyloid or tau pathology [3,17,23], but have been previously shown to have AD/ADRD-relevant neuronal pathologies driven by APOE4 expression, often appearing in an age-dependent fashion [31,61]. For each genotype, RNA was isolated from hemi-brains of mice at 12 MO (*n* = 6 per genotype) and 18 MO (*n* = 5 per genotype) and utilized for RNA-seq [32,46]. Sequences were interrogated using the DREAM pipeline [58] to identify DEGs between 12 MO and 18 MO mice within each genotype. The majority of DEGs identified were protein-coding (APOE2 92.57%, APOE3 88.92%, and APOE4 94.22%).

APOE2 mice had 5111 DEGs between 12 MO and 18 MO mice (Figure 1A), with 2344 DEGs upregulated by age and 2767 DEGs downregulated by age. APOE3 mice had 3069 DEGs between 12 MO and 18 MO mice (Figure 1B), with 1663 DEGs upregulated by age and 1406 DEGs downregulated by age. APOE4 mice had 6466 DEGs between 12 MO and 18 MO mice (Figure 1C), with 3122 DEGs upregulated by age and 3344 DEGs downregulated by age.

Comparing changes by age, large overlaps were observed between genotypes (Figure 2A), with 1610 DEGs altered from 12 MO to 18 MO mice in all three genotypes. This suggests that these 1610 DEGs, which we termed “pan APOE”, are dependent on aging rather than APOE genotype. Additionally, 1890 DEGs were shared between aging in APOE2 and APOE4. Virtually all of these shared DEGs (1605/1610: 99.7% and 1887/1890: 99.8%) were changed in the same direction with age (Figure 2B,C). The few (seven DEGs) whose aging effect changed direction between genotypes are listed in Appendix A.

To understand how genes differentiated by age may affect cellular function, IPA was employed to interrogate biological pathways. Interestingly, when looking at aging between APOE genotypes, 12 of the top 20 canonical pathways were shared between each genotype, and all but three pathways were significantly changed in every genotype per age comparison (Figure 3A). Using the 1610 pan APOE DEGs, the top 20 IPA canonical pathways were incorporated in the 27 pathways listed in Figure 3. Only four pathways changed direction between genotypes. For example, “Signaling by ROBO Receptors” was downregulated in APOE2 with aging but upregulated in APOE3 and APOE4 with aging. “Class I MHC mediated antigen processing and presentation” was upregulated in APOE3 but downregulated in APOE2 and APOE4 mice. The majority of the top 20 pathways were relevant to brain function, including the neurodegenerative pathway “Huntington’s Disease Signaling” (Figure 3A). A total of 469 common DEGs were found across aging in APOE2, APOE3, and APOE4 mice. The top 120 common DEGs were mechanistic drivers in at least 12 of the 27 canonical pathways (Figure 3A). These are represented as a circle plot (Figure 3B), depicting overlap between common driver DEGs and top IPA pathways. APOE4 exhibited the highest number of unique DEGs within these pathways (Appendix A). APOE4-unique DEGs also contributed to complex protein–protein interaction (PPI) networks (Appendix A), including those indicative of oxidative stress and metabolic dysfunction, likely adding to the impact of aging. Aggregating these results, aging is a larger driver of transcriptomic changes in the brain than APOE genotype in the absence of β-amyloid or tau pathology, with the APOE4 allele exacerbating the aging program compared to APOE2 and APOE3.

To study changes in pathways altered by aging specific to each APOE allele, analysis was conducted using DEGs uniquely altered by genotype. A curated list of the top relevant pathways altered by aging was generated for each genotype. APOE2, which is protective against neurodegeneration [11,12,62], had several lipid pathways downregulated during aging, including “Regulation of Lipid Metabolism by PPARalpha” and “Superpathway of Inositol Phosphate Compounds” (Figure 4), although a broad trend for APOE2-specific pathways was not discernible. In APOE3 mice, several signaling pathways, including “RHOGDI Signaling” and “S100 Family Signaling Pathway”, were altered by aging (Figure 5). DEGs unique to APOE4 correlate with increased risk of neurodegeneration [9,11], revealed the largest number of pathways specific to synaptic and neuronal signaling (Figure 6), with most being upregulated, including “Synaptic Long-Term Potentiation”, “Synaptogenesis Signaling Pathway”, and “Serotonin Receptor Signaling”. Notably, APOE4 was the only genotype that had enough unique DEGs altered from 12 MO to 18 MO to independently activate a neurodegenerative pathway (e.g., “Huntington’s Disease Signaling”) as a top pathway, with more than 6-fold significance compared to APOE2- or APOE3-unique DEGs.

KEGG was also used to interrogate top pathways changed by aging between APOE genotypes (Table 1). A total of 58 pathways were altered in APOE2, 19 pathways in APOE3, and 63 pathways in APOE4. As with IPA, the majority of pathways changed by aging in APOE3 were common with APOE2 and APOE4, yielding 18/19 (94.7%) overlapping pathways. A total of 39 pathways were shared between APOE2 and APOE4 age comparisons (9 shown in Table 1), likely being driven by the 1890 aging DEGs common only between APOE2 and APOE4. Among pathways shared between all genotypes, six were related to neurodegeneration, including “Alzheimer’s Disease” and “Huntington’s Disease”. There were not enough DEGs altered by age specifically in APOE2 and APOE3 mice to identify significant KEGG pathways. A total of 18 KEGG pathways were identified through aging DEGs unique to APOE4. Of these, three were shared between pathways that were significant when all DEGs in either APOE2, APOE3, or APOE4 were interrogated (Table 1), including “Pathways of Neurodegeneration-Multiple Disease”. An additional nine pathways from unique APOE4 DEGs overlapped with pathways from all APOE2 DEGs (e.g., not just unique APOE2 DEGs changed by aging). Therefore, six pathways altered by unique APOE4 DEGs were specific to this genotype.

GO analysis was employed to evaluate biological processes altered by aging between genotypes. Similar to IPA and KEGG, pan APOE DEGs accounted for the majority of biological processes enriched for each genotype (Figure 7A). Accordingly, GO processes were identified using DEGs altered in aging unique to each genotype, termed “unique APOE2”, “unique APOE3”, and “unique APOE4” (Figure 7B). Unique APOE4 processes were similar in distribution to GO processes from pan APOE DEGs. Processes categorized in autophagy and synaptic pathways found in unique APOE4 and pan APOE assessments were compared and contrasted to unique APOE2 DEGs and unique APOE3 DEGs. To account for the larger number of DEGs altered by aging unique to APOE4 compared to APOE2 or APOE3, the percentage of each category was assessed (Figure 7C–F). Both in terms of total number of processes and percentage of total processes, pan APOE and unique APOE4 were similar (Figure 7B,C,F). Synaptic processes were parsed into subcategories (Figure 8, Appendix A). There were few synaptic processes identified by “unique APOE2” or “unique APOE3” aging DEGs. These identified processes were largely associated with synaptic vesicle or synaptic transmission and function, with only one synaptic “other” process in “unique APOE3”. APOE4 aging DEGs were involved in synaptic processes spanning all subcategories (Figure 8), including 61 processes identified by pan APOE DEGs and 93 processes identified in “unique APOE4” DEGs. The largest subcategory was synaptic transmission and function. Therefore, pan APOE and unique APOE4 DEGs were most prominent in this assessment of transcriptomic changes in APOE targeted replacement mice.

## 4. Discussion

Mice with homozygous humanized targeted replacement APOE genes were utilized to interrogate DEGs and biological pathways from 12 MO and 18 MO cohorts. A large proportion of DEGs differentiated by age were common among all genotypes. Therefore, we postulate aging has a greater effect on changes in gene expression than APOE genotype. Independent studies support this contention, revealing significant interactions between age, APOE genotype, and sex, with age having the greatest impact [10]. To assess generalizability of the discovered aging ApoE4 genes in the present report, we compared DEGs to a publicly available dataset (GEO: GSE212343) from Lee et al. (2023) [39]. Although utilizing similar targeted replacement APOE4 mice (e.g., Taconic female humanized ApoE4 model), the whole brain bulk RNA-seq dataset from Lee et al. [39] revealed fewer DEGs than the present report, which may have been due to a variety of factors, including the number of samples utilized and other technical and/or methodological considerations. Despite experimental differences between studies, ~32.2% of the DEGs from Lee et al. [39] overlapped with the aging DEGs in APOE4 mice from 12 MO to 18 MO mice, with >81% showing convergent gene expression (e.g., changing in the same direction: either upregulation or downregulation). This initial cross-study comparison provides an interesting validation of a subset of aging APOE4 genes found in the present assessment and provides the impetus to continue comparing and contrasting multi-‘omics study results from humanized APOE mice for future mechanistic studies and potentially translational findings with therapeutic implications.

A total of 1610 DEGs were commonly differentiated between 12 MO versus 18 MO mice among all genotypes, which we termed pan APOE DEGs. This amounts to 52.5% of all APOE3 DEGs being pan APOE DEGs, with 24.0% of DEGs unique to APOE3. Likewise, only 23.5% of APOE2 DEGs and 41.1% of APOE4 DEGs were unique to respective genotypes (Figure 2). While the percentage was larger in APOE4, this was expected due to the greater overall number of APOE4 DEGs. We propose this is due to APOE4 exacerbating the effects of aging (e.g., Figure 3B and Appendix A). Overall, the subcategories of biological processes in GO analysis of the pan APOE DEGs more closely resembled unique APOE4 processes than unique APOE2 or unique APOE3 processes, including synaptic pathways (Figure 8). As age is the greatest risk factor for AD [33] and APOE4 is the largest genetic risk factor for late-onset AD [5,6], we posit APOE4 increases aging sequelae, rather than APOE4 directly influencing AD pathology. This is commensurate with the hypothesis that APOE plays a larger role in the context of age-related cognitive impairment, rather than solely serving as a risk factor for AD/ADRD [34,63].

The present results suggest APOE4 contributes to β-amyloid burden, independent of cognitive decline [19]. Our hypothesis is supported by APOE4 increasing the risk of several age-related disorders, including cardiovascular disease, type 2 diabetes mellitus [64], LBD [20,21,27], TDP-43 pathology severity [25,26], and cognitive decline in late Parkinson’s disease [65,66]. Conversely, APOE2 may be protective against aging, while APOE3 remains a neutral allele. APOE2 is protective in the context of AD [11,12,22]. APOE2 is also protective in other age-related diseases, including cerebral amyloid angiopathy [16], atherosclerosis [67], and LBD [27]. APOE2 carriers may develop a dementing illness 10 years later than APOE4 carriers [68] and have higher longevity, independent of AD pathology [62].

Via bioinformatic inquiry, we explored APOE genotype-specific pathways. Although IPA analysis covers a wide range of biologically relevant pathways beyond brain-themed pathways, the top 20 most dysregulated pathways in all genotypes are relevant to brain function (Figure 3). The largest group of dysregulated pathways was in the context of aging, including protein synthesis (e.g., “Eukaryotic Translation Initiation”, “RNA Polymerase II Transcription”, and “Processing of Caped Intron-Containing Pre-mRNA”, among others). KEGG analysis also indicated that “RNA polymerase”, “Ribosome”, and “Protein Processing in Endoplasmic Reticulum” are altered by aging, suggesting aging drives changes through alterations in transcription and translation. Alterations to metabolic pathways were also observed in aging, including “Oxidative Phosphorylation”, “Mitochondrial Dysfunction”, and “Respiratory Electron Transport” via IPA analysis (Figure 3) and “Oxidative Phosphorylation” via KEGG analysis (Table 1). These pathways are also significantly dysregulated across the progression of AD [49,69,70], with the APOE4 allele exacerbating MetS [30].

There were more alterations in gene expression by age specific to APOE4 compared to APOE2 or APOE3. APOE4 had the most DEGs (6466) from 12 MO to 18 MO mice and the greatest percentage of unique DEGs (41.1%) (Figure 2). Aging resulted in dysregulation of synaptic pathways across all analyses, including downregulation of “Synaptogenesis Signaling Pathway” in IPA, “Synaptic Vesicle Cycle” in KEGG, and 61 pathways with GO analysis (Figure 3 and Figure 7B, Table 1). This is expected, as APOE is localized in the synaptic milieu [71,72]. Aging DEGs unique to APOE4 also showed dysregulation of synaptic pathways. IPA displayed dysregulation of synaptic pathways in the top 20 pathways unique to APOE4 (Figure 6), including “Synaptic Long-Term Potentiation” and “Synaptogenesis Signaling Pathway”. Among the six KEGG pathways unique to APOE4, “Dopaminergic Signaling” was dysregulated (Table 1). GO analysis showed 93 synaptic biological processes specific to APOE4, compared to 5 and 8 biological processes in APOE2 and APOE3, respectively (Appendix A). There were more synaptic processes unique to APOE4 than those common between all genotypes, with the most in the “Transmission and Function” category (Figure 8).

Profound synaptic pathway dysfunction in the context of APOE4 is consistent with literature that indicates APOE4 decreases synaptic plasticity [72], synaptic density [73], and reduces mitochondrial capacity in astrocytes [74,75]. APOE4-targeted replacement mice, similar to those used in this study, have reduced spine density [74] compared to APOE2 or APOE3 mice at multiple timepoints. APOE4 mice also show impaired memory, as measured by the Morris water maze [76,77]. Similar to pathways identified herein, APOE4 mice have decreased mitochondrial function, ATP production, and mitochondrial density [75], as well as disrupted receptor recycling and reduced synaptic plasticity [73]. In addition to mouse models, hiPSC-derived neurons demonstrate APOE4 disrupts synaptic formation and neurotransmission pathways [40], decreases synaptic integrity [78], and alters neuronal excitability [35,79].

Limitations of the present study include a lack of understanding of how APOE drives downstream effects within the brain, especially in the context of hallmark AD/ADRD pathology (e.g., β-amyloid and tau), which is beyond the scope of this investigation. The lack of male mice in the 18 MO cohort is a potential confound and will be targeted in future studies. Importantly, the experimental design of the present study included aged females, which is translationally relevant as the ApoE AD/ADRD risk phenotype is stronger in females [10,13]. Thus, we were able to interrogate a high-risk group in humanized APOE murine models in the context of aging. Notably, no sex differences were identified in the 12 MO cohort or previous studies on these humanized APOE mouse models [32,46]. Further information on the relationship(s) between aging and APOE genotype can be gained from continuing to interrogate in vivo specific cell types (e.g., pyramidal neurons in hippocampus and cortical regions underlying memory and executive function) as well as interneurons and non-neuronal cell types (e.g., astrocytes and microglia) in vulnerable brain regions using single-cell or spatial transcriptomic profiling [39]. Limitations of bulk RNA-seq include loss of resolution for spatial and/or cell-specific drivers. This is a noted caveat of our initial study employing humanized APOE allele murine hemi-brain homogenates as the input sources. Studies have begun to compare profiling of bulk RNA-seq to single-cell populations in a subset of non-neuronal phenotypes [39]. To date, investigations of aging in the context of APOE alleles have limited resolution for neuronal changes or spatially derived network alterations. As such, single-population neuronal and/or spatial-derived validation is key for future studies in these well-established murine models.

## 5. Conclusions

The present findings demonstrate age has a greater effect on gene dysregulation than APOE genotype. Moreover, we show differential gene expression and mechanistic pathways underlying alterations caused by aging in the context of the APOE allele. Few changes are seen specific to APOE3, indicating that this common allele is neutral. APOE4 exacerbates aging effects, notably in synaptic pathways. We also identified underlying gene expression changes directly impacting synaptic dysregulation in the context of the APOE4 allele in this in vivo model system. Taken together, these data provide mechanistic and therapeutic insight into both aging and AD.

## Figures and Tables

**Figure 1 brainsci-15-01117-f001:**
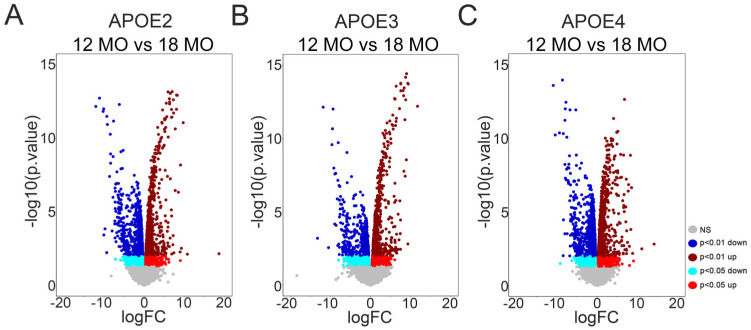
Distribution of DEGs comparing 12 MO to 18 MO mice. DEGs were identified between 12 MO and 18 MO mice homozygous for APOE2 (**A**), APOE3 (**B**), and APOE4 (**C**). Volcano plots show the distribution by log-fold change (logFC). Blue represents DEGs downregulated with age (light blue: *p*-value < 0.05, dark blue: *p*-value < 0.01) and red represents DEGs upregulated with age (red: *p*-value < 0.05, dark red: *p*-value < 0.01).

**Figure 2 brainsci-15-01117-f002:**
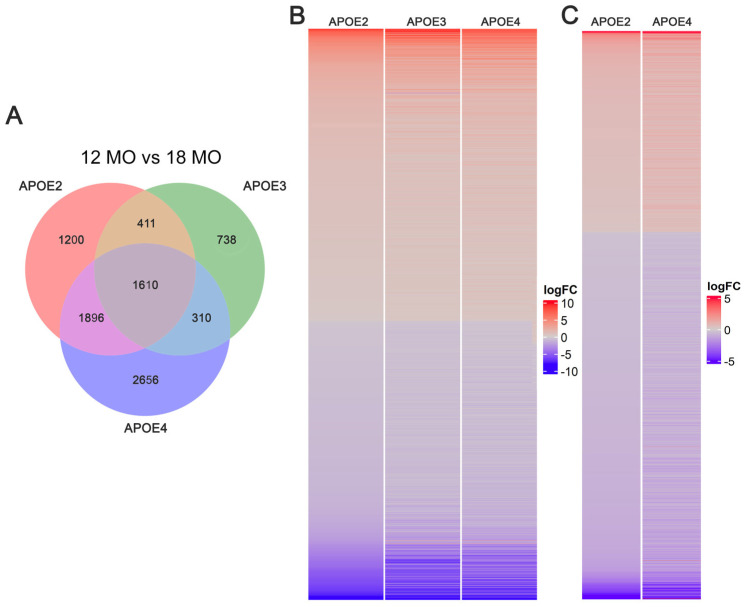
Comparison of shared DEGs between genotypes. (**A**) Venn diagram depicts overlap of aging DEGs between APOE2 (pink), APOE3 (green), and APOE4 (purple). The number of shared and unique DEGs are listed. (**B**) Heatmap shows the logFC of the 1610 DEGs shared between all 3 genotypes. All but 7 DEGs changed in the same direction. (**C**) Heatmap of 1896 DEGs shared between APOE2 and APOE4 illustrating DEGs changing in the same direction. Even though more DEGs were shared between APOE2 and APOE4 than all 3 genotypes, the range of logFC was smaller.

**Figure 3 brainsci-15-01117-f003:**
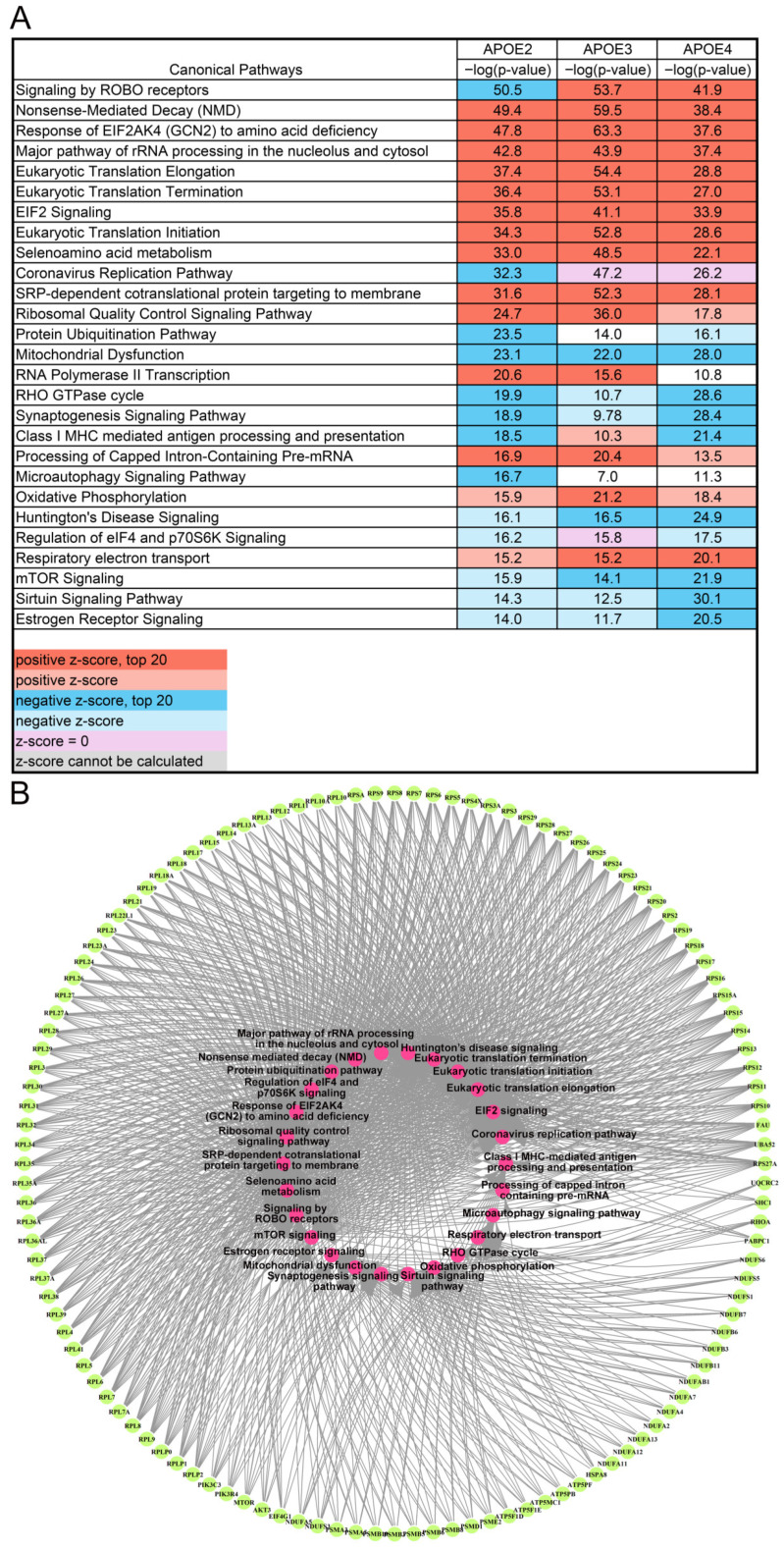
(**A**) Top 20 canonical pathways via IPA across genotypes. IPA generated a list of all canonical pathways significantly differentiated between 12 MO and 18 MO timepoints. The −log(*p*-values) of the top 20 pathways for each genotype were combined into one list. Orange depicts pathways upregulated (positive z-score) and blue depicts pathways downregulated, with shading indicating the top 20. Purple indicates that the z-score was 0 and grey indicates that a z-score could not be calculated. (**B**) Circle plot illustrating the top DEGs driving canonical pathway dysregulation, shown in light green, with the corresponding pathways in the center (pink) and grey arrows drawn between individual DEGs and pathways of interest.

**Figure 4 brainsci-15-01117-f004:**
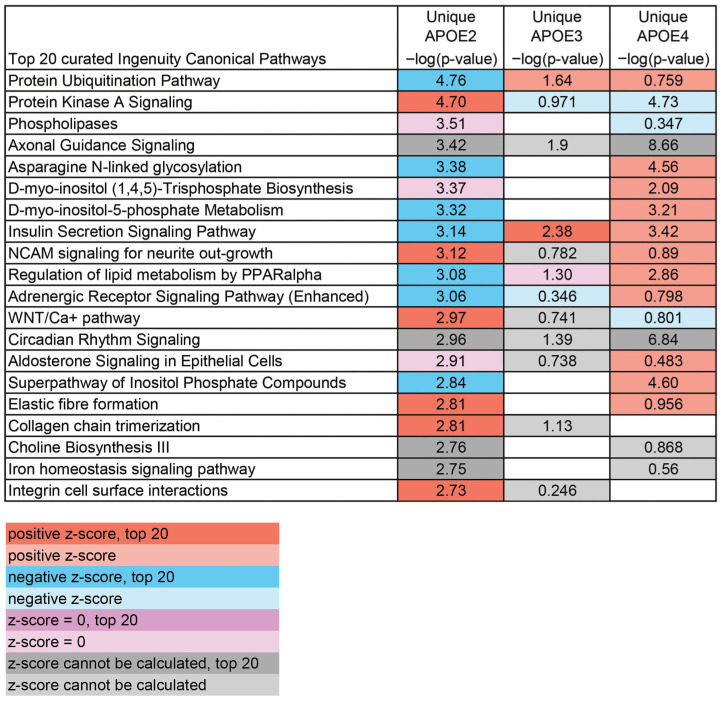
Curated top 20 list of canonical pathways via IPA unique to APOE2. A curated list of the top 20 pathways from DEGs unique to APOE2 was created and compared to pathways altered by aging in APOE3 and APOE4. Orange depicts pathways upregulated (positive z-score) and blue depicts pathways downregulated, with shading indicating the top 20. Purple indicates that the z-score was 0 and grey indicates that a z-score could not be calculated. Shading indicates only 1 top 20 pathway shared with APOE2.

**Figure 5 brainsci-15-01117-f005:**
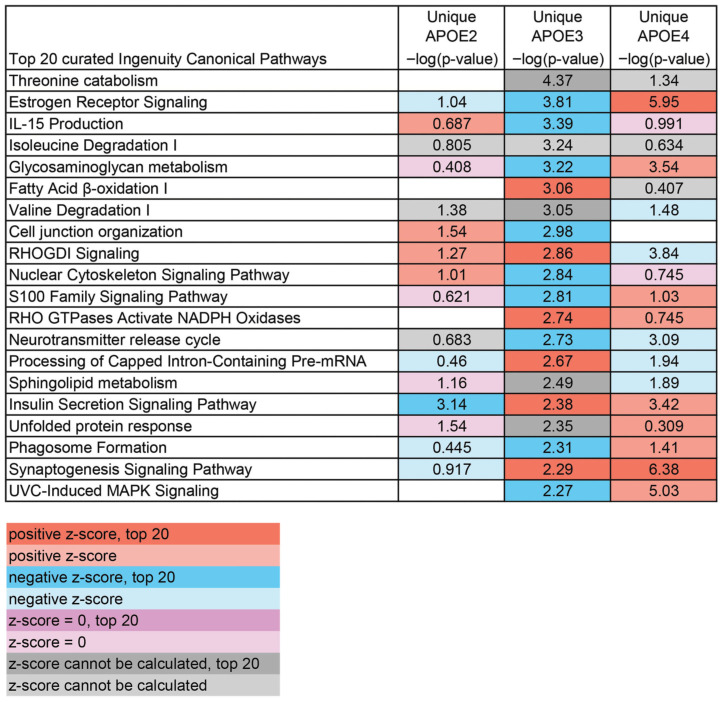
Curated top 20 list of canonical pathways via IPA unique to APOE3. A curated list of the top 20 pathways from DEGs unique to APOE3 was generated and compared to the pathways altered by aging in APOE2 and APOE4. Orange depicts pathways upregulated (positive z-score) and blue depicts pathways downregulated, with shading indicating the top 20. Purple indicates that the z-score was 0 and grey indicates that a z-score could not be calculated. Shading indicates only 3 top 20 pathways shared with APOE3.

**Figure 6 brainsci-15-01117-f006:**
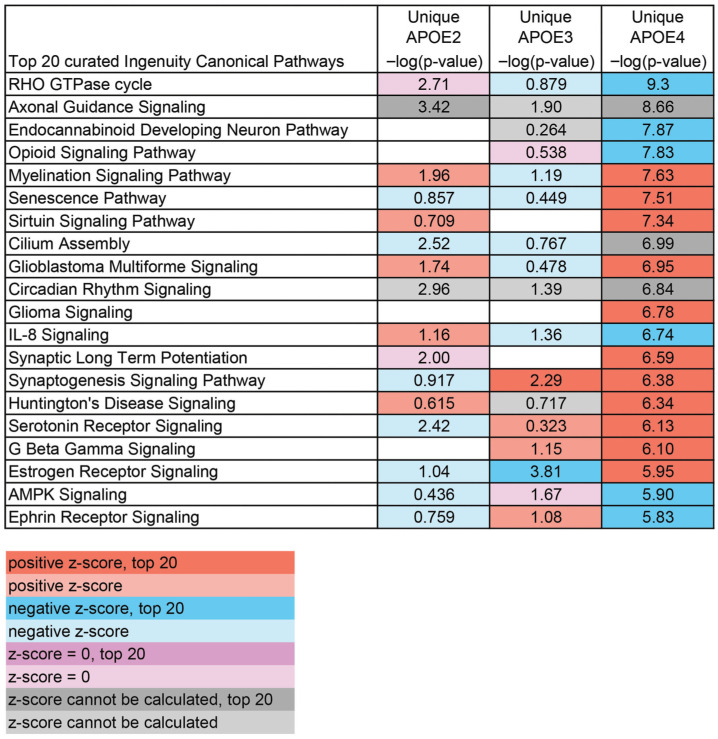
Curated top 20 list of canonical pathways via IPA unique to APOE4. A curated list of the top 20 pathways from DEGs unique to APOE4 was created and compared to the pathways altered by aging in APOE2 and APOE3. Orange depicts pathways upregulated (positive z-score) and blue depicts pathways downregulated, with shading indicating the top 20. Purple indicates that the z-score was 0 and grey indicates that a z-score could not be calculated. Shading indicates only 2 top 20 pathways shared with APOE4.

**Figure 7 brainsci-15-01117-f007:**
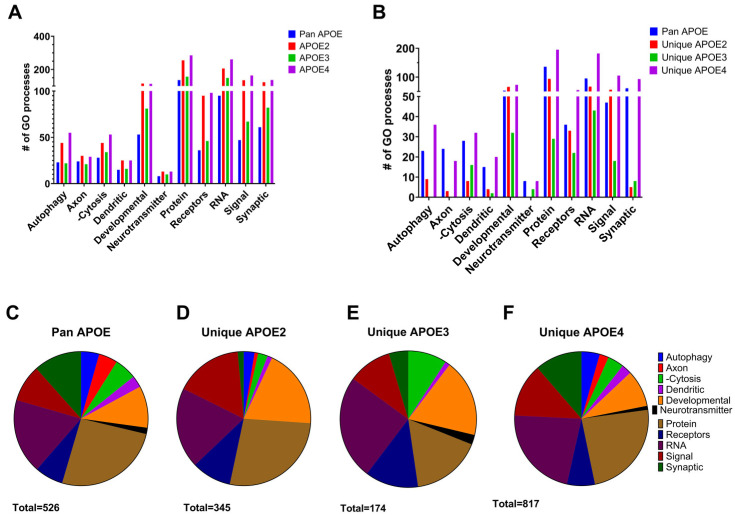
Curated biological pathways via GO analysis. Biological processes identified via GO analysis were categorized into autophagy, axon, -cytosis (exocytosis, endocytosis, and phagocytosis), dendritic, developmental (only those present in the CNS), neurotransmitter, protein, receptors, RNA, signal transduction, and synaptic. Bar graph in (**A**) shows the analysis for all the DEGs in APOE2, APOE3, APOE4, and pan APOE (with logFC values from APOE4). Bar graph (**B**) shows the analysis for pan APOE and aging DEGs unique to APOE2, APOE3, and APOE4. Pie graphs show the percentage of each category shown in **B** (**C**–**F**). Values are shown for aging DEGs in pan APOE (**C**), unique APOE2 (**D**), unique APOE3 (**E**), and unique APOE4 (**F**).

**Figure 8 brainsci-15-01117-f008:**
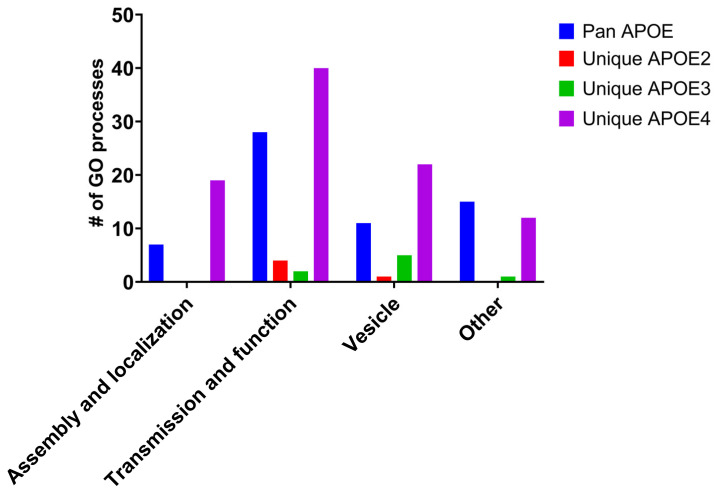
Synaptic subcategories within GO analysis. The synaptic category from the GO analysis was parsed into subcategories. Bar graph shows the total number of biological processes in each category for comparisons of unique DEGs in APOE2, APOE3, and APOE4. Additionally, processes were compared to pan APOE DEGs with logFC values from APOE4.

**Table 1 brainsci-15-01117-t001:** List of pathways via KEGG analysis. KEGG generated a list of canonical pathways significantly differentiated between 12 MO and 18 MO. The −log(*p*-value) of pathways altered with all DEGs in APOE2, APOE3, and APOE4 are listed. DEGs unique to APOE4 generated additional KEGG pathways. There were not enough aging DEGs unique to APOE2 or APOE3 to significantly alter any pathways in KEGG. An additional 30 pathways were changed with all aging DEGs in APOE2 and APOE4, but not in APOE3, which are not shown.

Description	APOEZ −log(*p*-Value)	APOE3 −log(*p*-Value)	APOE4 −log(*p*-Value)	APOE4 Unique −log(*p*-Value)
Parkinson’s disease	13.89	17.67	23.63	
Huntington’s disease	18.45	17.95	23.08	
Alzheimer’s disease	12.97	16.04	21.58	
Amyotrophic lateral sclerosis	15.74	16.42	18.28	
Prion disease	10.36	16.19	17.68	
Retrograde endocannabinoid signaling	8.50	5.03	14.14	
Spinocerebellar ataxia	8.92	7.39	13.63	
Thermogenesis	11.07	11.93	13.44	
Autophagy–animal	8.30	3.42	11.16	
Oxidative phosphorylation	11.97	15.45	10.93	
Ubiquitin-mediated proteolysis	7.50	4.58	8.08	
Ribosome	8.69	22.91	5.69	
Proteasome	3.22	3.54	4.05	
Synaptic vesicle cycle	2.74	2.58	3.75	
RNA polymerase	3.23	2.64	3.08	
Pathways of neurodegeneration–multiple diseases	14.32	16.36	21.97	2.87
Mitophagy–animal	5.30	2.76	12.08	3.05
Protein processing in endoplasmic reticulum	6.54	2.45	6.24	2.87
Axon guidance	7.35		12.31	6.10
Endocytosis	5.42		10.93	4.57
Gap junction	4.77		10.61	2.84
Long-term potentiation	3.93		8.36	2.51
Glutamatergic synapse	3.37		8.72	3.71
Inositol phosphate metabolism	2.55		2.95	2.57
Choline metabolism in cancer	2.42		5.80	2.49
Rap1 signaling pathway	2.17		5.62	4.13
Adherens junction	2.05		5.09	3.67
Dopaminergic synapse			8.07	3.91
Phospholipase D signaling pathway			5.71	3.51
EGFR tyrosine kinase inhibitor resistance			5.38	2.45
Amphetamine addiction			5.09	4.75
Cocaine addiction			3.37	4.29
MAPK signaling pathway			3.12	3.90
Nucleotide excision repair	3.88	3.35		

## Data Availability

The datasets supporting the conclusions of this article are available as follows. RNA-seq data analyzed within this study are available from GEO with 12 MO data accessible through GEO Series accession number GSE188267 (https://www.ncbi.nlm.nih.gov/geo/query/acc.cgi?acc=GSE188267, accessed on 10 October 2025) and 18 MO data accessible through GEO Series accession number GSE234711 (https://www.ncbi.nlm.nih.gov/geo/query/acc.cgi?acc=GSE234711, accessed on 10 October 2025) or from the corresponding author upon request.

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
