# Peer review of "Aging, Rather than Genotype, Is the Principal Contributor to Differential Gene Expression Within Targeted Replacement APOE2, APOE3, and APOE4 Mouse Brain"

_brainsci, 2025, doi:10.3390/brainsci15101117_

Round 1
Reviewer 1 Report
Comments and Suggestions for Authors
The manuscript looks interesting, but some modifications are recommended.
- In introduction, references 2 and 4 for the distribution of APOE should be updated.
- Why the authors specifically used the left hemibrains of the mice?
- Which alignment method was used in STAR aligner?
- A subheading about statistical analyses used should be added to the methodology section.
- I suggest constructing protein-protein interaction network between the identified top 20 canonical pathways via IPA across genotypes and APE alleles.
- Some figures can combine together.
- The discussion lacks perspectives and solutions to confirm in other settings the results described in the manuscript. This must be included.
Author Response
Reviewer #1
We thank the Reviewer for a thorough review and insightful comments. We emended the manuscript accordingly in a point-by-point fashion below.
- “References 2 and 4 for the distribution of APOE should be updated.”
Response: Corrected with current references (e.g., lines 49, 52-54, 522-530). A total of 15 new references were added to the text.
- “Why the authors specifically used the left hemibrains of the mice.”
Response: Right hemibrains were used for biochemistry as described previously in Peng et al., 2019, 2024 (lines 113-114).
- “Which alignment method was used in STAR aligner?”
Response: We are sorry for any confusion. We revised the text to indicate STAR aligner was used to generate a genome index and to align trimmed reads to the indexed murine reference genome (lines 130-132).
- “A subheading about statistical analyses used should be added.”
Response: Based on this inquiry, we added a subheading ‘2.4. Statistical Analysis and Bioinformatics’ (line 133).
- “Constructing protein-protein interaction network between the identified top 20 canonical pathways via IPA.”
Response: Thank you for requesting this information. We revised the manuscript accordingly by including Supplementary Figure 1 where we generated a STRING analysis in Cytoscape to identify PPIs from 368 DEGs (out of a total of 441) with associated text (lines 204-211). An additional author, Kyrillos W. Ibrahim, M.S., assisted with the analysis and graphics output.
- “Some figures can combine”.
Response: We politely disagree with this comment. The seven multi-panel figures (now with an additional panel for Fig. 3B) are image- and text-laden. Condensing them would compromise readability in our view. We prefer to keep the 7 figure format with the 600 DPI resolution.
- “Discussion lacks perspectives and solutions to confirm in other settings.”
Response: Based on this inquiry, we emended the text in the Discussion to include additional externalities, limitations, and future solutions in the context of aging and APOE alleles (lines 451-454 and 461-469).
Reviewer 2 Report
Comments and Suggestions for Authors
The manuscript addresses an important and timely question regarding the relative impact of aging versus APOE genotype on brain transcriptomic changes in humanized APOE knock-in mice. That said, some clarifications, additional analyses, and improvements in presentation would strengthen the manuscript.
Major Comments
At 12 months, both sexes are represented, but the 18-month cohort includes only females. The absence of males at the later time point introduces a confound, especially given known sex differences in APOE-associated risk. The authors acknowledge this limitation, but further discussion of how this impacts the interpretation of results is warranted.
Interpretation of “Pan-APOE” DEGs: The 1,610 DEGs shared across genotypes are attributed mainly to aging. While convincing, it would be valuable to show whether these DEGs overlap with known age-related transcriptomic signatures from independent datasets (mouse or human). Such a comparison would enhance the translational significance.
Pathway Analyses
The manuscript uses IPA, KEGG, and GO. While informative, there is some redundancy in the presentation. A clearer hierarchy (e.g., presenting overlapping findings first, then genotype-specific divergences) would improve readability. It would also help to include a network-level visualization (e.g., Cytoscape or STRING) summarizing pan-APOE versus APOE4-specific pathways.The authors emphasize synaptic dysregulation in APOE4. Expanding the discussion to connect these transcriptomic changes with functional evidence (e.g., known synaptic impairments in APOE4 models, electrophysiology, or behavioral outcomes) would strengthen the biological interpretation.
Statistical Thresholds: DEGs were selected at p < 0.05 without FDR correction. This is lenient given the scale of RNA-seq data. The rationale should be better justified, and ideally, results at an adjusted threshold (e.g., FDR < 0.05) should be provided, at least in supplementary material.
Minor Comments
Consider in the introduction the study in different popolations by APOE factor risk; e.i. ( doi: 10.3390/brainsci14090908)
Methods: Please clarify if batch effects (sequencing lanes) were controlled in the DREAM/variancePartition model.
Discussion: Expand on limitations related to reliance on whole hemi-brain homogenates (loss of cell-type resolution). A short mention of potential future single-cell or spatial transcriptomic approaches would be valuable.
References: Some citations appear outdated or redundant. More recent large-scale transcriptomic aging/AD studies (2023–2025) should be added for context.
Author Response
Reviewer #2
We thank the Reviewer for helpful comments regarding our submission. We appreciate the queries and respond in a point-by-point fashion below.
Major points:
- “Absence of males at the later time point introduces a confound, especially given known sex differences in APOE-associated risk. The authors acknowledge this limitation, but further discussion.”
Response: Text in the Discussion was revised to include this limitation of the current study and provide a rationale for studying females with the intent of examining aged males as part of future studies (lines 450-454 and 461-469).
- “Valuable to show whether these DEGs overlap with known age-related transcriptomic signatures from independent datasets (mouse or human).”
Response: This is an interesting suggestion that we gave significant thought to. In response, we compared the present DEGs from the ApoE4 aging paradigm to a similar ApoE4 model aging comparison from the relatively recently published Lee et al., 2023 (publicly available GEO: GSE212343; Reference #39). Interestingly, the head-to-head comparison revealed ~32% of the DEGs from Lee et al., 2023 overlapped with the DEGs in our ApoE4 aging paradigm, and >81% showed convergent gene expression (e.g., in the same direction). Although not a multi-study dataset comparison, this keen suggestion that we followed through with provides proof-of-concept for at least partial overlap of aging APOE4 genes across studies and paradigms (lines 370-384). Exhaustive comparative pathway analyses are beyond the scope of this report, but could be envisioned using curated DEGs from several published studies to create a validated pan ApoE4 aging geneset.
- “A clearer hierarchy…would improve readability. It would also help to include a network-level visualization.”
Response: Thank you for this inquiry. We revised text and graphics output (Figure 3 and Supplementary Figure 1) in response to this request. As described in the response to Reviewer #1, comment 5, we organized data presentation for greater clarity, including presentation of PPI networks (Supplementary Figure 1, including depiction in STRING by Cytoscape) as well as providing network level assessment of DEGs as part of a new panel in Figure 3 (e.g., Fig. 3B; Figure Legend lines 246-261) describing the top DEGs driving the canonical pathway dysregulation (lines 204-211).
- “Expanding the discussion to connect these transcriptomic changes with functional evidence”
Response: Based on this critique, we revised the Discussion to include convergence of synaptic DEG alterations in APOE4 mice with functional changes in physiology and behavior (lines 439-446).
- “Statistical Thresholds: DEGs were selected at p < 0.05 without FDR correction.”
Response: There are many levels of stringency that can be selected for bioinformatic inquiry (e.g., ranging from highly lenient inclusion of all sequenced genes to highly stringent inclusion only DEGs that pass FDR at 0.1 and many ranges in between). We routinely choose the standard threshold of DEGs at p < 0.05 in our multi-‘omics studies to ensure there are sufficient DEGs for pathway analyses, including IPA (e.g., ideally 200-2,000 DEGs), KEGG, and GO as indicated in with References (References #47-49; line151).
Minor points:
- “Introduction the study in different popolations by APOE factor risk.”
Response: In response to this comment, the suggested meta-analysis reference (Reference #8) was added to the Introduction (lines 55-59).
- “Please clarify if batch effects (sequencing lanes) were controlled in the DREAM/variancePartition.”
Response: The were no batch effects in this study. All samples were assayed at the same time at NYUGSOM GTC. This is indicated in Section 2.3 (lines 125-127),
- “Limitations related to reliance on whole hemi-brain homogenates (loss of cell-type resolution). A short mention of potential future single-cell or spatial transcriptomic approaches would be valuable.”
Response: In response to this suggestion, additional Discussion text was added in the context of Reviewer #2, comment 2 (lines 461-469).
- “Some citations appear outdated or redundant.”
Response: Similar to the critique raised by Reviewer #1, comment 1, we revised citations throughout the text to be current and reflective of multi-‘omics studies in APOE mice (e.g., References 1-4, 7-8, 10, 24, 35, 39, and 48).
Round 2
Reviewer 2 Report
Comments and Suggestions for Authors
Thanks for consider my revision.